# MoS-VLA: A Vision-Language-Action Model with One-Shot Skill Adaptation

## Abstract

Vision-Language-Action (VLA) models trained on large robot datasets promise general-purpose, robust control across diverse domains and embodiments. However, existing approaches often fail out-of-the-box when deployed in novel environments, embodiments, or tasks. We introduce Mixture of Skills VLA (MoS-VLA), a framework that represents robot manipulation policies as linear combinations of a finite set of learned basis functions. During pretraining, MoS-VLA jointly learns these basis functions across datasets from the Open X-Embodiment project, producing a structured skill space. At test time, adapting to a new task requires only a single expert demonstration. The corresponding skill representation is then inferred via a lightweight convex optimization problem that minimizes the $L^1$ action error, without requiring gradient updates. This gradient-free adaptation incurs minimal overhead while enabling rapid instantiation of new skills. Empirically, MoS-VLA achieves lower action-prediction error on five out of five unseen datasets and succeeds in both simulation and real-robot tasks where a pretrained VLA model fails outright.

## 1 Introduction

Inspired by the success of large language models, modern robotics aims to achieve generalization and human-like performance through the use of internet-scale data and large, attention-based architectures. To this end, researchers have collected enormous datasets of robotic arm trajectories (Open X-Embodiment Collaboration et al., 2023) and trained so-called vision-language-action foundation models to map natural language task descriptions and state observations to robot actions (Kim et al., 2024; Octo Model Team et al., 2024; Brohan et al., 2023b;a; Ma et al., 2024). While these models achieve acceptable performance on in-distribution tasks and lab settings, they are so far brittle to out-of-distribution data. The key challenge is that there is no preexisting dataset of robotic demonstrations analogous to the enormous amount of natural language data present on the internet, and collecting data in robotics settings is expensive and difficult to scale. Thus, there is currently not enough data to achieve generalization through scale alone. Therefore, creating a generalist robotic policy inherently requires a more principled approach to transfer.

The current state-of-the-art approach to transfer is to finetune a pretrained vision-language-action model on a (relatively) small calibration dataset collected on the new task and/or lab setting. Often, this calibration dataset consists of 10s or 100s of expert demonstrations, and finetuning the models requires significant resources. Even parameter-efficient finetuning techniques (e.g., LoRA, QLora) (Han et al., 2024; Hu et al., 2022; Dettmers et al., 2023) inherently require significant resources as numerous forward and backward passes are required during gradient descent.

In contrast to finetuning, we present a gradient-free adaptation method based on function encoders. The function encoder algorithm learns a set of neural network basis functions to span a space of interest. In this case, the basis functions span the space of policies, where a given policy is specialized to a specific lab setting. After training, a new policy for a new lab setting is computed as a linear combination of the basis functions, where the coefficients are calculated as the solution to a least absolute error regression problem. Because the optimization space is small and well-structured, only a tiny calibration dataset consisting of one expert trajectory is needed. Furthermore, the least absolute error problem is just a linear program written in terms the basis function outputs. Therefore, no gradient calculations are needed and the model adaptation has the smallest possible overhead: only

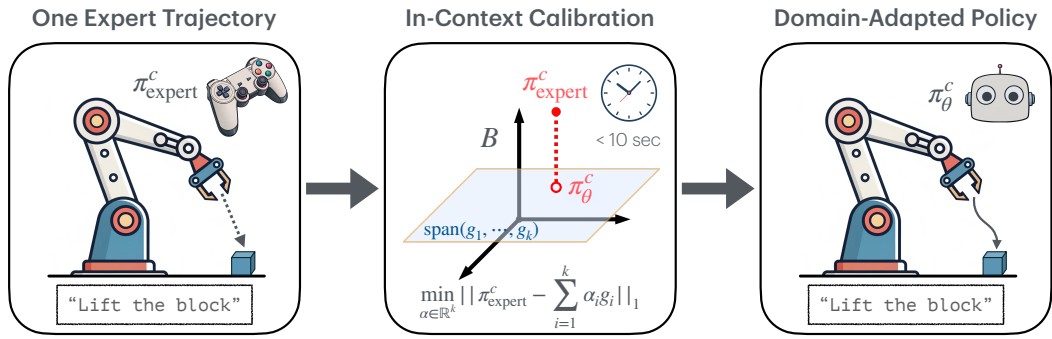

Figure 1: **In-context adaptation with function encoders.** (*Left*) A human expert collects one trajectory of data in the new context. (*Middle*) The model is calibrated by solving a small, least absolute error optimization problem. This process takes only a few seconds, even on an RTX 3090. (*Right*) The adapted model autonomously executes actions in the new environment.

forward passes are necessary. Lastly, due to the simplicity of least absolute error optimization, we demonstrate the ability to adapt to a new lab setting given one expert demonstration in only a few seconds, using only an RTX 3090 GPU.

We empirically verify our approach by building on top of OpenVLA (Kim et al., 2024) and training on the RT-X dataset (Open X-Embodiment Collaboration et al., 2023). We demonstrate that our approach achieves improved accuracy on the training and evaluation datasets. Additionally, we evaluate the policies on a Franka Emika Panda robot arm in unseen lab settings in both simulation and the real world. The pretrained OpenVLA model achieves a 0% success rate due to the domain gap. In contrast, after calibrating with one expert trajectory, our approach achieves a 70 to 100% success rate across all tasks, demonstrating in-context transfer.

**Main Contributions:**

- We introduce a one-shot, in-context adaptation method for vision-language-action models. Our method has a significantly reduced overhead relative to prior works because it does not require gradient computations after training.

- We are the first to demonstrate that the function encoder algorithm is applicable to billion-parameter architectures and mixed vision-language datasets.

- We empirically validate that the proposed method outperforms baselines both on datasets and in real-world experiments.

## 2 RELATED WORKS

**Foundation Models for Robotics** A recent paradigm for robotics to replicate the successful formula of language models, which is internet-scale data combined with billion-parameter attention-based architectures. The Open X-Embodiment dataset is a massive collection of robotic data designed to mimic a internet-scale training regime (Open X-Embodiment Collaboration et al., 2023). Numerous works train large transformer models using this data, which may be diffusion models (Octo Model Team et al., 2024) and/or auto-regressive (Kim et al., 2024), and tend to incorporate pretrained language and vision models such as Llama 2 (Touvron et al., 2023), Google-T5 (Raffel et al., 2020), SigLIP (Zhai et al., 2023), and DinoV2 (Oquab et al., 2024). Despite the magnitude of resources dedicated to these approaches, empirically they fail to generalize to out-of-distribution tasks and lab settings. Furthermore, given the expense of robotic data collection and the diminishing returns implied by the so-called scaling laws (Kaplan et al., 2020), robotic foundation models require the development of more sophisticated and principled methods of transferring to new tasks and settings. We present one such method based on transfer within the span of a set of neural network basis functions (Ingebrand et al., 2025).

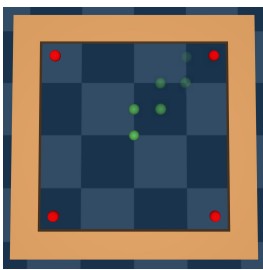 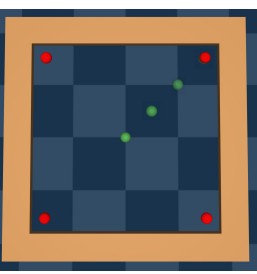

Figure 2: **Discrete vs. Continuous Skill Primitives.** *(Left)* A discrete policy restricted to "right" or "up" yields inefficient, Manhattan-style paths and cannot reach all goals. *(Right)* Continuous combinations of skills enable direct movement in any direction, allowing the robot to reach all goals.

**Adaptation and In-Context Learning** The principle method for adaptation is to finetune the foundation model on a relatively small dataset collected from the new setting and/or task (Kim et al., 2025). While this dataset is typically orders of magnitude smaller than the pretraining dataset, it may still consists of many trajectories collected over many tasks. Thus, there is still significant overhead in finetuning as these datasets must be collected for every new setting. Furthermore, given the size of foundation models, which are often billions of parameters, full-finetuning requires a powerful machine, often consisting of numerous GPUs. Parameter-efficient fine-tuning methods such as LoRA are used to reduce the finetuning overhead at the cost of expressivity and performance (Han et al., 2024; Hu et al., 2022; Wang et al., 2024). Nonetheless, any method with leverages gradient descent requires significant overhead, both in compute and in memory, due to the numerous forward and backward passes required for gradient descent.

Rather than computing gradient updates, a natural solution is to append expert demonstrations as another input to the foundation model (Fu et al., 2024), in the style of in-context learning for language models (Dong et al., 2024; Min et al., 2022). As is typical for transformers, this method is quadratic in both memory and compute with respect to the expert trajectory length. In contrast to this approach, our method treats the expert trajectory as a *set* of input-output pairs rather than a *sequence*, which removes the need to compute attention between all data pairs. Consequently, our method is linear with respect to the expert trajectory length during an initial calibration step, and constant thereafter.

**Skill Learning and Task Representations** Prior methods, especially from the reinforcement learning (RL) community, learn diverse skills and/or task representations. These methods often leverage information-theoretic proxy objectives such as cross entropy (Eysenbach et al., 2019; Hausman et al., 2018). Other approaches investigate how to best leverage pretrained skills for downstream tasks (Lim et al., 2019). Meta-RL aims to learn policies that are quickly adapted to new tasks (Lan et al., 2019), while zero-shot RL uses offline datasets to learn spaces of task solutions that do not require extra training at inference time (Touati & Ollivier, 2021; Ingebrand et al., 2024b). Our method is similar to these approaches in that we learn a space of policies, although we only leverage supervised learning techniques rather than dynamic programming. Additionally, our space of policies encodes different behaviors *and* different contexts due to the visual variations between lab settings. Lastly, mixture of experts is common technique for foundation models that creates sub-networks specialized to certain tasks. Each task is routed to a different sub-network via a learned gating mechanism (Cai et al., 2025; Puigcerver et al., 2024). The intention is to lower the computational complexity of inference, since only a subset of the total parameters are active. As a consequence, only one or a few experts are active for any given input. In contrast to learning a finite *set* of experts, our method trains basis functions to work in tandem, creating a *space* of policies. See Figure 2 for a thought experiment motivating the use of continuous policy spaces over discrete sets.

## 3 BACKGROUND

### 3.1 FUNCTION ENCODERS

The function encoder algorithm (Ingebrand et al., 2025) learns basis functions for an arbitrary function space. Given a Hilbert space $\mathcal{H} = \{f : X \to Y\}$ with an inner product $\langle \cdot, \cdot \rangle_{\mathcal{H}}$ and an induced norm $|| \cdot ||_{\mathcal{H}}$, function encoders learn a set of neural network basis functions $\{g_1, ..., g_k\}$, where $k$ is a hyper-parameter, to span the space $\mathcal{H}$. Any function $f \in \mathcal{H}$ is represented as a linear combination of the basis functions, $f = \sum_{i=1}^{k} \alpha_i g_i$, where $\alpha \in \mathbb{R}^k$. Furthermore, given the

basis functions and a function $f$, $\alpha$ is calculated as the solution to the least-squares minimization, $\alpha := \arg\min_{\alpha \in \mathbb{R}^k} ||f - \sum_{i=1}^k \alpha_i g_i||_{\mathcal{H}}^2$.

The training procedure requires a set of datasets, $\mathcal{D} = \{D_1, ..., D_n\}$, where each dataset $D_i = \{x_j, f_i(x_j)\}_{j=1}^m$ is a set of input-output pairs for a function $f_i \in \mathcal{H}$. The training procedure uses gradient descent. At each gradient step, the coefficients $\alpha$ for function $f_i$ are calculated by solving the least-squares minimization. Then, the function is approximated as a linear combination of the learned basis, and the loss is the error of this approximation. See Ingebrand et al. (2025) for more information.

In contrast to the typical function encoder definitions, we will leverage Banach spaces and $L^1$ optimization rather than $L^2$ optimization. We define the coefficients as

$$\alpha := \arg\min_{\alpha \in \mathbb{R}^k} ||f - \sum_{i=1}^k \alpha_i g_i||_1. \tag{1}$$

We use least absolute error instead of least squares because the RTX dataset is noisy. Ignoring outliers and emphasizing precise control is crucial for good performance[1]. However, the only practical difference is the optimization procedure to compute the coefficients, where we solve a linear program instead of an unconstrained quadratic program. The rest of the algorithm remains unchanged. See Appendix E for a comparison.

## 4 METHOD

To enable in-context transfer with minimal overhead and minimal calibration data, we leverage the function encoder algorithm with a few modifications. For simplicity, we build upon the OpenVLA foundation model (Kim et al., 2024), although we highlight that the following approach can be applied to any model architecture. The method section is outlined as follows. In Section 4.1, we formally define in-context adaptation. In Section 4.2, we define the core technical idea of our approach. In Section 4.3, we describe the necessary architecture changes to create $k$ basis functions. In Section 4.4, we outline the training procedure.

### 4.1 PROBLEM FORMULATION

We consider the following setup. The robot observes its environment through RGB images of a fixed size, denoted by the set $I$. Tasks are described in natural language and drawn from a set $T$. For simplicity, both $I$ and $T$ are assumed finite, representing the (large but bounded) number of images and tasks the robot may feasibly encounter. A robot's state is defined as a pair $s \in S = I \times T$, consisting of an image and a task description. The robot's action space is denoted $A \subset \mathbb{R}^m$, where each output dimension corresponds to a robot end effector's translation, rotation, etc[2]. A trajectory $\tau = \{(s_t, a_t)\}_{t=1}^T$ is a sequence of states and actions of length $T$. Each trajectory is collected under a fixed but unobserved context $c \in C$, which captures factors such as lighting, camera position, robot morphology, decision rate, etc. We write $\tau^c$ to highlight that a trajectory was collected under context $c$, and we emphasize that the context is relevant for decision making but it is not extractable from the state directly. Lastly, we assume there exists an expert policy $\pi_{exp} : S \times C \to A$, which is by definition optimal with respect to some objective function. The expert policy, typically a human, is assumed to take the context into consideration as part of its decision making. We seek to train a policy $\pi_\theta : S \to A$ such that $\pi_\theta(s) = \pi_{exp}(s, c)$ for all states and contexts. In other words, we seek to train a model to clone the expert policy. However, this objective is ill-posed because the expert observes the context but our learned model does not. We argue that the ill-posed nature of this problem explains why VLA models fail to generalize despite the large amount of training data, and later we will argue how in-context adaptation methods address this problem.

Next, we will define the data assumptions. Let $D^c = \{\tau_1^c, \tau_2^c, ...\}$ be a set of expert trajectories collected in one lab setting. Note that we assume that the context is fixed for a given lab setting, as

---

[1]Prior work has also argued in favor of $L^1$ losses for this problem setting (Kim et al., 2025). Our work concurs with that finding, even in the context of function encoders.

[2]Prior works instead choose to learn a distribution over a discretized action space (Kim et al., 2024). See Appendix D for a discussion

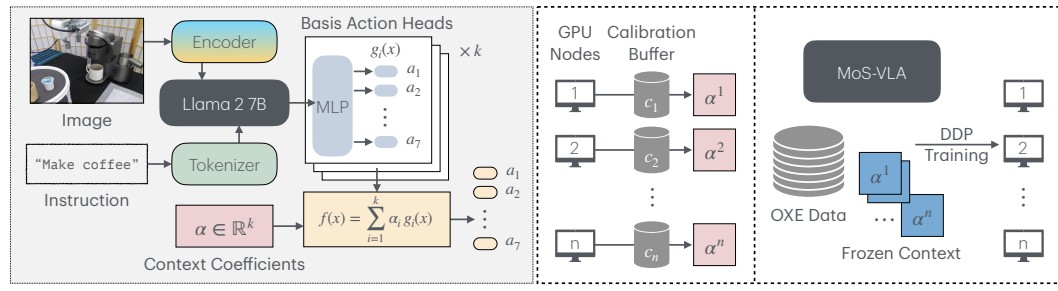

| MoS-VLA Model Architecture | Context Calibration | Parameter Update |

Figure 3: **The MoS-VLA Architecture and Training Pipeline**. *(Left)* MoS-VLA builds on the OpenVLA backbone where images and instructions are processed by a Llama 2 model. The language modeling head is replaced with $k$ basis action heads. The action predictions are linearly combined via vector operations as described in Section 3.1. *(Right)* During training, we iterate between context estimation and parameter updates. We utilize parallelism in both stages: during calibration, each training node computes and broadcasts the coefficient only for the datasets it hosts. During model updates, the context coefficients are held constant and trainable weights are adjusted with DDP.

robot implementation details are unlikely to change between trajectories. Let $\mathcal{D} = \{D^{c_1}, D^{c_2}, ...\}$ be a set of such datasets. For the purposes of this paper, $\mathcal{D}$ is the Open X-embodiment dataset, as it consists of data from multiple labs, where each lab's dataset consists of multiple trajectories.

The in-context adaptation problem consists of two stages: First, there is an offline training period for $\pi_\theta$ using $\mathcal{D}$, where training time is not a concern. Second, there is an online adaptation phase. The trained model $\pi_\theta$ is provided with one additional trajectory $\tau^c_{exp}$ collected by the expert policy under a context $c$. Typically, $c$ is an unseen context that was not present in the offline training data set. Then $\pi_\theta$ outputs an action $a$ for a new state $s$, and should reproduce the behavior of $\pi_{exp}$. Thus, $\pi_\theta$ should use the additional information provided by $\tau^c_{exp}$ to improve its accuracy on this new context. In this second phase, adaptation time should be minimized, ideally to seconds or less.

### 4.2 MAIN IDEA

To design a policy that adapts to the context given the trajectory $\tau^c_{exp}$, we will formulate the set of context-dependent expert policies as a function space, and use $\tau^c_{exp}$ to identify a function within this space. Formally, let $\pi^c_{exp}(\cdot) = \pi_{exp}(\cdot, c)$ be the expert policy conditioned on a context $c$, where this notation makes the dependence on the context implicit. Consider the set of such policies $\Pi_{exp} = \{\pi^c_{exp} | c \in \mathcal{C}\}$. We define the vector operations for this space as the standard element-wise operations, and the norm is defined as $||f||_1 := \int_S \sum_{i=1}^{\dim A} |f_i(s)| ds$. Thus, we equip $\Pi_{exp}$ with a Banach space structure (see Oden & Demkowicz (2018)). Then, by using the function encoder algorithm, we can learn a set of neural network basis functions $\{g_1, ..., g_k\}$ to span $\Pi_{exp}$, where any policy $\pi^c_{exp} = \sum_{i=1}^{k} \alpha^c_i g_i$. During the online adaptation phase, we compute $\alpha^c \in \mathbb{R}^k$ by a least absolute error optimization problem. Once we have these coefficients $\alpha^c$, we may approximate $\pi^c_{exp}$ for any new state $s$ using a linear combination of the basis functions.

A key advantage of this approach is its computational simplicity during the online phase. The function encoder adapts to new contexts via the least absolute error calculation, which has two simple steps. First, we must forward pass the basis functions on each state $s$ in $\tau^c_{exp}$. Note that we can forward pass all states in parallel. Second, we solve the least absolute error problem (1), which is effectively just a linear program solvable in a negligible amount of time. The function encoder's overhead for in-context adaptation is minimal. Crucially, no backpropagation is required. Thus, our method adapts to new contexts in the same amount of time required to forward pass the model, which is only seconds even for relatively weak GPUs. Furthermore, once we have computed the coefficients $\alpha^c$, we output actions as a linear combination of the basis functions. Therefore, the computational and memory costs after calibration is constant with respect to the expert trajectory length.

## 4.3 ARCHITECTURE

To make OpenVLA amenable to the function encoder algorithm, we require $k$ separate output actions conditioned on the state, where $k$ is a hyper-parameter. Furthermore, the input has variable length depending on the task description, and so each basis function must accept variable length sequences as input, i.e., each basis function must use attention. While prior works have used non-traditional architectures such as neural ODEs for the basis functions (Ingebrand et al., 2024a), we are the first to leverage transformers as basis functions for the function encoder algorithm, and the first to demonstrate that the algorithm scales to function spaces with images and natural language as inputs.

Due to the large number of parameters in OpenVLA, creating $k$ fully independent copies is infeasible. Instead, we leverage a single OpenVLA network body with multiple output heads. Specifically, we remove the Llama 2 output head and replace it with $k$ randomly initialized output heads. In addition, we utilize the parallel decoding technique in OpenVLA-OFT to model all action dimensions simultaneously. The output of each head acts as a separate basis function, and each outputs an action. See Figure 3. Note that the original OpenVLA implementation uses the Llama 2 head rather then training a new one from scratch. Thus, it outputs a distribution over Llama 2's vocab, which contains 32,000 tokens, and the least-used 256 tokens are replaced with action tokens. In contrast, each basis function only outputs scalar action values directly. Thus, despite the extra action heads, MoS-VLA uses only 0.2% more parameters than OpenVLA. In general, the overhead in terms of parameter count is small, and this same procedure can be applied to other foundation models as well.

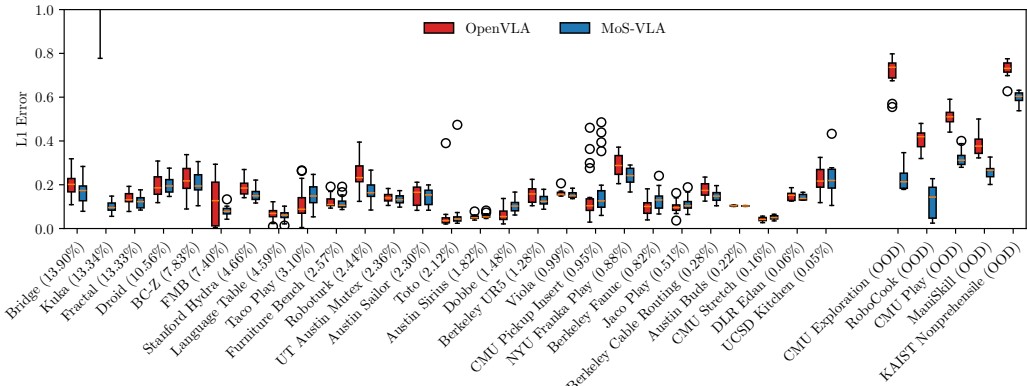

Figure 4: **Validation Action $L^1$ Error Across Train and OOD Datasets**. We train MoS-VLA on the same Open X-Embodiment Magic Soup Plus data mixture as OpenVLA. Each subset is annotated with its proportion in the training set. Datasets marked as OOD are unseen during training. MoS-VLA, calibrated on only 512 training examples, achieves lower error than OpenVLA on 18 of 27 training subsets and on all 5 OOD datasets.

## 4.4 TRAINING PROCEDURE

We start with the pretrained OpenVLA model. After removing the Llama 2 output head and initializing the $k$ basis function heads, it is clear we must train these parameters directly. For the rest of the OpenVLA parameters, we may choose to either finetune them using LoRA or to fully finetune them. For the sake of computational efficiency, we use LoRA for the results in this paper. Training progresses according to the modified function encoder algorithm using Banach definitions and $L^1$ optimization. See Appendix B for the pseudocode.

Starting with a pretrained OpenVLA is a design choice to reduce the length of the offline training period. In principle, we could instead start with pretrained vision and language models, as other VLA models do (Bjorck et al., 2025), and train the basis functions from there. Doing so may improve the performance, as the pretrained OpenVLA model may trap the model's performance at a local optimum given that OpenVLA was trained on classification instead of regression. We leave this direction for future work.

## 5 EXPERIMENTS

**Implementation Details.**    We train 16 basis functions by finetuning OpenVLA on the RTX dataset as described in Section 4. We utilize 32 distributed compute nodes, each of which has a GH200, using data parallelism (DDP) (Li et al., 2020). We train for a total of 5000 gradient steps, which takes approximately 24 hours. In order to fairly compare against OpenVLA, we use the same *Open-X Magic Soup Plus* data mixture from Kim et al. (2024) with a global batch size of 320. We use an Adam optimizer with a learning rate of $1e^{-4}$ and a warmup of 10 steps.

**Scalable Skill Calibration.**    In the standard function encoder algorithm, the coefficients for each context are calculated every training step using a batch of training data, and gradients are back-propagated through this computation. This procedure requires a reasonably large batch size to ensure stable estimation of the context coefficients, which is practically infeasible for VLA training. For the sake of memory efficiency, and scalability to distributed training of a large model, we instead maintain a calibration buffer with a capacity of 512 training samples for each dataset. During DDP training, the calibration buffers are evenly spread across computation nodes. In our case, with 32 nodes and 27 datasets, the first 27 nodes each hosts a unique dataset. We compute a new set of coefficients every 16 gradient steps: nodes hosting the calibration buffer compute and broadcast the coefficients across all nodes. After each calibration, the coefficients are held constant, and gradients are not back-propagated through this process. We find that this implementation is memory efficient, since it does not have to solve a linear program each gradient step, but it does not negatively impact performance. Additionally, it allows us to easily maintain the data sampling ratio from the data mixture, whereas the standard function encoder approach would not. We solve the linear program corresponding to (1) via CVXPY (Agrawal et al., 2019).

### 5.1 RESULTS

**$L^1$ Error on Datasets.**    After training, we evaluate our model against state-of-the-art VLA models (Kim et al., 2024; Physical Intelligence et al., 2025) on both in-distribution and out-of-distribution datasets, with results shown in Figure 4. Specifically, we compute the context coefficients for each subset from 512 random data points from its training split. While OpenVLA achieves similar performance on in-distribution datasets, it is significantly worse when evaluated out-of-distribution. We attribute this behavior to a phenomenon we term *mixed-context overfitting*. This occurs when an over-parameterized model, lacking mechanisms to distinguish between contexts, is trained on a mixed-context dataset. In such cases, the model learns a single function to approximate many contexts simultaneously, resulting in sharp, unstable outputs across contexts. Note this type of over-fitting occurs even if the data itself is noise-free. In contrast, using the function encoder algorithm mitigates this type of overfitting because there is effectively a different model for each context. See Appendix A for a visualization of this phenomenon in a 1D setting.

Additionally, we leverage the collected data to visualize the learned function space. For each task across all lab settings, we compute the corresponding skill coefficients and apply dimensionality reduction techniques to reveal the structure of the learned landscape. As shown in Figure 6, the coefficients form clusters that align with lab settings, supporting our hypothesis that each lab environment is effectively captured by a unique function. Notably, three Austin datasets—treated as separate during training—naturally cluster together, reflecting their underlying similarity in lab conditions. These results highlight the potential of the learned skill space to provide interpretability and to expose similarities across datasets.

| Method | Simulation ($m = 20$) | | On-robot ($m = 10$) | | |
| | Lift Block | Open Door | Reach | Lift Block | Insert Pen |
|---|---|---|---|---|---|
| **OpenVLA** | 0% | 0% | 0% | 0% | 0% |
| $\pi_{0.5}$ | 0% | 0% | 0% | 0% | 0% |
| **ICRT** | 0% | 0% | 0% | 0% | 0% |
| **MoS-VLA (1-shot)** | 70% | 75% | 100% | 100% | 100% |

Table 1: Success rate comparison between OpenVLA and MoS-VLA.

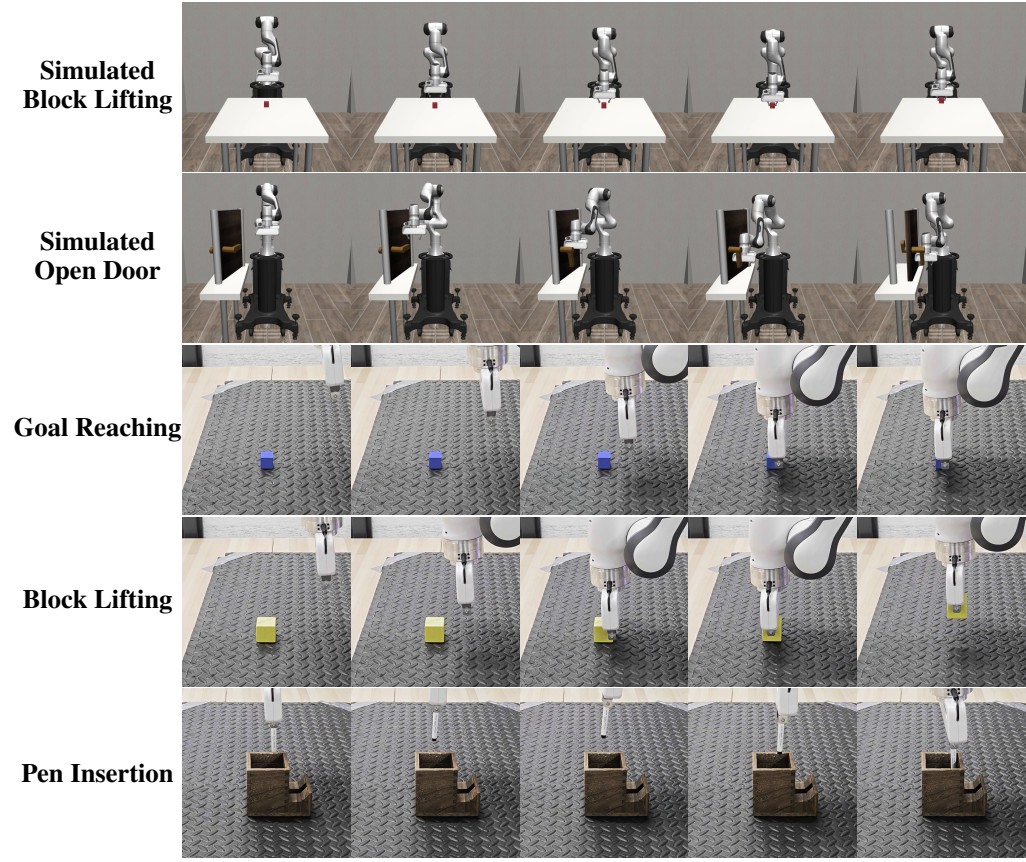

Figure 5: **Illustration of simulated and real robot tasks.** MoS-VLA rollouts across five tasks in both simulated and on-robot experiments. Having never seen these tasks during training, MoS-VLA needs only one demonstration for gradient-free adaptation. Note that the objective for goal reaching is to hover above the block.

**Policy Performance.** To further evaluate our model, we run simulated and on-robot experiments with the Franka Emika Panda robot arm. For simulation, we use two tasks from Robosuite (Zhu et al., 2020): block lifting and door opening. For on-robot experiments, we solve three tasks: goal reaching, block lifting, and pen insertion. In all experiments, the model is calibrated using one expert trajectory only. Detailed descriptions of the test domains are as follows:

(1) **Simulated Block Lifting:** A red block is placed on a table beneath the robot gripper randomly to the left or right. The robot is allowed to move sideways or up/down to grasp and lift the block away from the table. (2) **Simulated Door Opening:** A door is placed to the right of the robot about the same height as its base. The robot gripper is allowed to move sideways and up/down to push and grab the handle. (3) **Goal Reaching:** A blue block is placed at a fixed location on the mat in front of the robot. The robot can move in $x, y$ and $z$ directions to reach and hover above the block. (4) **Block Lifting:** A larger yellow block is placed in front of the robot. The robot can move in $x, y$ and $z$ directions to grasp and lift the block. (5) **Pen Insertion:** The robot is initialized with a pen in its gripper. It is allowed to move sideways and up/down to insert the pen into the pen holder. Because the VLA models receive only a single front-facing RGB view and no proprioceptive state inputs, we simplify tasks 1,2, and 5 by disabling forward/backward motion to reduce the difficulty of depth estimation.

Our robot evaluation results are shown in Table 1. We find that the base OpenVLA model and $\pi_{0.5}$ are brittle to unseen environments and completely fail both in simulation and on the real robot. This is because neither the simulation nor our lab setting appear in the training set, and these models struggles to generalize. We also evaluate an in-context learning method (Fu et al., 2025) and find

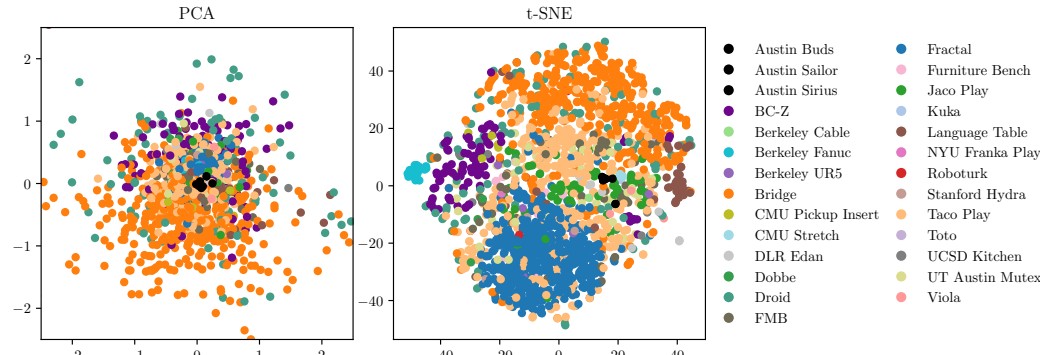

Figure 6: **Visualization of the Coefficients.** Coefficients for each task and lab setting are computed from expert trajectories, then visualized using PCA and t-SNE. We observe that tasks from the same lab setting generally cluster together in the learned function space. The PCA visualization shows that the first two principle components only poorly distinguish between datasets, suggesting that more than two basis functions are needed. We also plot three Austin datasets in black because they are known to have similar lab setups. In both PCA and t-SNE, these datasets appear as one cluster, demonstrating that the coefficient space captures this similarity.

that it too fails to generalize, even when provided the calibration data. In contrast, by calibrating with a single expert demonstration, MoS-VLA adapts to the new settings.

While MoS-VLA demonstrates clear advantages in our test environments, its success remains limited to relatively short-horizon tasks. Extending the framework to long-horizon problems will likely require additional mechanisms for temporal abstraction or hierarchical skill composition. Moreover, environments with higher stochasticity—such as the simulated block lifting task—may demand more than one demonstration to achieve consistent success.

## 6 CONCLUSION

We introduced a method for training VLA models to form a structured skill space that enables one-shot domain adaptation. Adaptation is achieved via least absolute error optimization, requiring no gradient calculations. This design makes the approach especially suitable for hardware-constrained settings where fine-tuning is impractical. Empirically, we demonstrate that our method improves performance on unseen robot experiments, outperforming a pretrained OpenVLA model that lacks domain transfer capability.

This work represents a first step toward gradient-free, in-context adaptation of robot foundation models. We argue that the need to fine-tune robot policies for every new domain remains a major barrier to the deployment of general-purpose robotics. Future directions include exploring alternative functional spaces, training schemes, and architectural choices. By building expandable skill spaces, our paradigm opens the door to VLA models that can rapidly acquire new skills without erasing prior knowledge.

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

## A    MIXED-CONTEXT OVERFITTING

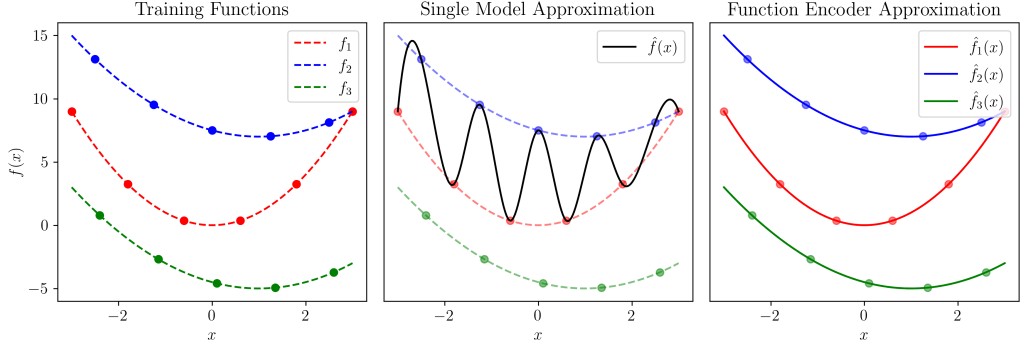

Figure 7: **Visualization of the Mixed-Context Overfitting Phenomenon.** *(Left)* The two training functions, $f_1$ and $f_2$, are shown in red and blue along with the training data points. The third function, $f_3$, is held out. *(Middle)* A single, over-parameterized function approximator is fit to the training data. *(Right)* A set of basis functions is fit to the training data. After calibrating, the basis functions may approximate all three functions accurately.

In Figure A, we visualize how overfitting is especially problematic for over-parameterized approximators in a mixed-context (mixed-function) setting. Suppose we have a dataset consisting of two functions, $f_1$ and $f_2$, along with some data points from each. If we fit a single function approximator, e.g., a neural network, it may perfectly fit the training data so long as there is no input $x$ common to both functions' datasets. However, this fit generalizes poorly, and cannot be calibrated to fit $f_3$ without further training.

On the other hand, a function encoder approximation computes coefficients for any given function, and so it effectively learns a space of functions rather than a single approximation. Assuming that the basis functions have been trained to span the training functions, each training function will be well-represented as a linear combination of the basis. After calibration, the function encoder even fits the unseen function so long as it is within the span of the basis. See Ingebrand et al. (2025) for more details on generalization to out-of-distribution functions. In summary, the function encoder avoids mixed-context overfitting by learning a separate function for each context.

## B  ALGORITHM PSEUDOCODE

The algorithm is similar to the function encoder algorithm in Ingebrand et al. (2025), where the primary difference is the minimization problem solved to compute the coefficients. We additionally regularize the Gram matrix to be close to identity, i.e., the basis functions are approximately orthonormal. For completeness, we include the pseudocode in Algorithm 1. Note that the inner product is defined as $\langle f, g \rangle = \int_{\mathcal{X}} f(x)^T g(x) dx$, which is approximated using the trajectory datasets.

---

**Algorithm 1** Modified Function Encoder Algorithm (LAE)

---

  **given** expert policy space $\Pi_{exp}$, learning rate $\alpha$
  Initialize basis $\{g_1, \ldots, g_k\}$ with parameters $\theta$
  **while** not converged **do**
    **for all** $\pi_{exp}^c \in \Pi_{exp}$ **do**
      $\alpha^c = \arg\min_{\alpha^c \in \mathbb{R}^k} ||\pi_{exp}^c - \sum_{i=1}^k \alpha_i^c g_i||_1$
      $\hat{\pi}_{exp}^c = \sum_{i=1}^k \alpha_i^c g_i$
    **end for**
    $L \leftarrow \sum_{\pi_{exp}^c \in \Pi_{exp}} ||\pi_{exp}^c - \hat{\pi}_{exp}^c||_1$
    $L_{reg} \leftarrow \sum_{i=1}^k \sum_{j=1}^k (\langle g_i, g_j \rangle - \delta_{ij})^2$
    $\theta \leftarrow \theta - \alpha \nabla_\theta (L + L_{reg})$
  **end while**
  **return** $\{g_1, \ldots, g_k\}$

---

## C  FUNCTION ENCODER ACTION HEAD ARCHITECTURE

The function encoder action head maps backbone features into $k$ basis action predictions. It is implemented as a two-block MLPResNet, following the parallel decoding strategy from OpenVLA-OFT. The input is a feature vector of dimension $28672$ (flattened from $4096 \times 7$). This vector is projected into a hidden dimension of $4096$, processed through two residual MLP blocks, and then mapped to the output dimension $k \times d_a$, where the action dimension is $d_a = 7$. Thus, the output corresponds to $k$ basis functions, each producing a 7-dimensional action vector. Formally, given input $h \in \mathbb{R}^{28672}$, the head produces $B(h) \in \mathbb{R}^{k \times 7}$, where each row $B_i(h)$ denotes the prediction of the $i$-th basis function.

## D  DISCRETE VS. CONTINUOUS ACTIONS

OpenVLA learns a discrete distribution over a discretized version of the action space, and it generates the actions autoregressively Kim et al. (2024). This design decision was likely motivated by two factors. First, this allows OpenVLA to use the Llama 2 model without making any architectural changes. Second, it naturally allows for complex action distributions, such as stochastic bang-bang control. In contrast, a continuous action model is unlikely to learn bang-bang control and is more likely to learn the mean of this behavior. We choose to focus on continuous actions because this more naturally aligns with the function encoder algorithm. However, it is also possible to create a discrete distribution function encoder. See Ingebrand et al. (2025), Appendix C.

## E  ABLATIONS

We perform ablations to evaluate how sensitive our model is to its hyper-parameters, namely the number of basis functions, the number of data points used to calibrate the model, and the distance function definition. Due to compute limitations, we only run one seed for each hyper parameter value. Unless otherwise specified, all ablations use 16 basis functions, 512 calibration data points, and $L^1$ distance.

**Number of Basis Functions**   We vary the number of basis functions and retrain the model. We report the training results in Table 2. We observe that the algorithm is relatively insensitive to the

number of basis functions beyond 8, suggesting that even 8 basis functions is sufficient to span the RTX dataset.

| Basis Functions | ManiSkill | CMU Exploration | KAIST Nonprehensile | Robocook | CMU Play | Mean |
|---|---|---|---|---|---|---|
| 4 | $0.2846 \pm 0.1135$ | $0.3517 \pm 0.1703$ | $0.5785 \pm 0.1210$ | $0.2398 \pm 0.1014$ | $0.3297 \pm 0.1009$ | $0.3569 \pm 0.1214$ |
| 8 | $0.2535 \pm 0.1275$ | $0.2290 \pm 0.2529$ | $0.5798 \pm 0.1275$ | $0.1295 \pm 0.1631$ | $0.3021 \pm 0.1831$ | $0.2988 \pm 0.1708$ |
| 16 | $0.2570 \pm 0.1383$ | $0.2308 \pm 0.2496$ | $0.5844 \pm 0.1316$ | $0.1297 \pm 0.1588$ | $0.3057 \pm 0.1782$ | $0.3015 \pm 0.1713$ |
| 32 | $0.2646 \pm 0.1402$ | $0.2639 \pm 0.2298$ | $0.5901 \pm 0.1345$ | $0.1457 \pm 0.1493$ | $0.3135 \pm 0.1539$ | $0.3155 \pm 0.1616$ |

Table 2: Validation $L^1$ error for $K = 4, 8, 16, 32$ across OOD datasets (mean $\pm$ std).

**Number of Calibration Data Points** We vary the size of the calibration buffer and retrain the model. We plot the results in Figure 8. As expected, we find that a larger calibration buffer is more stable and converges faster, but it costs more compute.

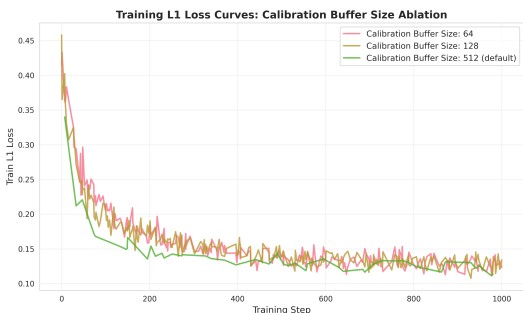

Figure 8: $L^1$ Loss during training for various calibration buffer sizes. The results indicate that a larger calibration buffer converges faster and more stably.

$L^1$ **vs** $L^2$ We compare $L^1$ distance definitions against $L^2$ distance definitions. $L^1$ requires solving a linear program, while $L^2$ has a closed form solution via least squares. We report the results in Table 3, which indicate that $L^1$ performs slightly better in this setting.

| Method | ManiSkill | CMU Exploration | KAIST Nonprehensile | Robocook | CMU Play | Mean |
|---|---|---|---|---|---|---|
| $L^1$ | $0.2570 \pm 0.1383$ | $0.2308 \pm 0.2496$ | $0.5844 \pm 0.1316$ | $0.1297 \pm 0.1588$ | $0.3057 \pm 0.1782$ | $0.3015 \pm 0.1713$ |
| $L^2$ | $0.2652 \pm 0.1130$ | $0.2473 \pm 0.2050$ | $0.5803 \pm 0.1230$ | $0.1549 \pm 0.1338$ | $0.3321 \pm 0.0997$ | $0.3159 \pm 0.1349$ |

Table 3: Validation $L^1$ accuracy comparison between $L^1$ and $L^2$ models, reported as mean $\pm$ standard deviation for each dataset.

# F COMPARISONS TO OTHER ADAPTATION METHODS

| Method | N = 32 | | | N = 128 | | | N = 512 | | | |
|---|---|---|---|---|---|---|---|---|---|---|
| | $L^1$ | Time | VRAM | $L^1$ | Time | VRAM | $L^1$ | Time | VRAM | Fits? |
| MoS-VLA (Ours) | $0.3410 \pm 0.1894$ | 1.5 | 17.0 | $0.3158 \pm 0.1776$ | 2.6 | 17.1 | $0.3015 \pm 0.1713$ | 13.1 | 17.6 | ✓ |
| LoRA Discrete | $0.3329 \pm 0.1936$ | 169.0 | 34.2 | $0.3226 \pm 0.1848$ | 122.4 | 35.4 | $0.3075 \pm 0.1763$ | 108.2 | 35.2 | ✗ |
| LM Head Only | $0.4728 \pm 0.1901$ | 81.0 | 19.5 | $0.4384 \pm 0.1891$ | 78.4 | 19.8 | $0.4165 \pm 0.1662$ | 78.8 | 19.8 | ✓ |
| OpenVLA-OFT | $0.3269 \pm 0.1534$ | 280.5 | 42.2 | $0.3128 \pm 0.1391$ | 287.7 | 50.3 | $0.2879 \pm 0.1345$ | 302.5 | 50.1 | ✗ |

Table 4: Comparison across methods for different sample sizes (N).

The key benefit of our method is the low overhead in terms of compute and memory at inference time. To demonstrate this, we compare our method against LoRA finetuning baselines on OpenVLA and OpenVLA-OFT. We plot the results in Table 4. Note that the last column indicates if the method fits in the memory of the RTX 3090, which has 24 GB of RAM. We also ablate how sensitive each method is to the number of calibration data points.

The results indicate that only our method and LoRA finetuning the projection head of an OpenVLA model fits in the 3090. Of the two, our method performs significantly better in terms of $L^1$ accuracy. Furthermore, our method takes substantially less time and RAM than all other methods, highlighting the efficiency of our approach. In terms of accuracy, finetuning OpenVLA-OFT performs slightly better than our approach, but requires minutes to calibrate, whereas our method requires only seconds. Furthermore, OpenVLA-OFT requires substantially more RAM to calibrate than our method does, and it would not fit on the provided hardware.

