# OpenReview forum: "MoS-VLA: A Vision-Language-Action Model with One-Shot Skill Adaptation"
_ICLR.cc/2026/Conference — Submitted to ICLR 2026_

### Official Review · Reviewer_qder · 2025-10-29

**Soundness:** 3
**Presentation:** 2
**Contribution:** 2
**Rating:** 4
**Confidence:** 5

**Summary:**

The paper MoS-VLA: A Vision-Language-Action Model with One-Shot Skill Adaptation introduces a method that helps robots quickly adjust to new environments and tasks using only one demonstration. Instead of retraining large models through expensive fine-tuning, MoS-VLA represents robot control policies as combinations of smaller learned skills, called basis functions. During training, these basis functions are learned from diverse robot datasets to form a flexible skill space. When facing a new task, the model calculates a simple optimization to find the right mix of skills, without any gradient updates. This allows fast and efficient adaptation, even on regular hardware. Experiments show that MoS-VLA achieves lower prediction errors on multiple unseen datasets and succeeds in both simulation and real-world robot tasks where standard models fail. The work demonstrates that one-shot adaptation can effectively transfer learned abilities to new conditions and helps bridge the gap between large-scale training and practical deployment.

**Strengths:**

The paper proposes *MoS-VLA*, a vision-language-action (VLA) framework that achieves one-shot skill adaptation through a gradient-free convex optimization procedure. The motivation is strong—reducing the high computational overhead of fine-tuning large VLA models for new environments is a relevant and timely challenge in embodied robotics. The method is clearly described, mathematically grounded in the function encoder formalism, and demonstrates an elegant way to represent policies as combinations of learned basis functions. The architecture and training pipeline are systematically presented, and both simulated and real-robot experiments confirm the feasibility of lightweight adaptation. The paper is well written, conceptually coherent, and provides promising insights into how structured skill spaces can support efficient policy transfer in large embodied models, even though its empirical validation remains limited in scale and complexity.

**Weaknesses:**

1.The real world tests (block lifting, goal reaching, pen insertion) are very short-horizon and low-complexity. These setups mainly assess spatial calibration, not compositional reasoning or long-horizon manipulation, such simple tasks can also be solved by small sclae reinforcement learning policy, making it unclear whether the proposed one-shot adaptation method scales to realistic long-horizon or contact-rich tasks, in which the VLA outperform recent tranditional policy learning framework.

2.Only a single robot platform (Franka Panda) and three real-world tasks are reported, each adapted from demonstration by one expert trajectory. There is no systematic analysis of robustness to in-the-wild scenarios such as noisy demonstrations, viewpoint shifts, or sensor noise, which limits confidence in the generality of the claimed one-shot transfer.

3.The paper compares only to OpenVLA and not to other lightweight adaptation strategies (e.g., LoRA-based finetuning, prompt-conditioning, diffusion-based adapters, or in-context imitation frameworks). Without these, the improvement magnitude is difficult to interpret.

4.Key hyperparameters such as the number of basis functions, choice of L1 vs. L2 objective, and calibration buffer size are fixed without detailed justification. No experiments isolate how these factors affect adaptation speed or accuracy.

5.The central algorithm directly builds on function encoder algorithm Ingebrand et al. (2025) with modest architectural adjustments (using transformer basis heads). The contribution therefore feels incremental rather than theoretically new.

Overall In summary, while the proposed approach is seemingly interesting, the limited and small-scale experiments fail to convincingly demonstrate its effectiveness compared with the rapidly advancing state-of-the-art in VLA.

**Questions:**

1.The key contribution for this paper is inherently introducing a one-shot adaptation method in the VLA domain that require only one expert demonstration for new task adaptation. As I understand, this method should be framework agnostic, did author attempt using other open-sourced VLA models for testing (e.g., pi0)? Or this method only valid for OpenVLA?

2.Can the authors provide larger-scale real-robot evaluations or report robustness tests (e.g., noisy demonstrations, camera shifts, or unseen lighting conditions) to validate one-shot adaptation under real-world unstructured environments? It is best to provide the complete real-world demonstration videos via an anonymous link.

3.Why are recent lightweight adaptation baselines, such as LoRA-based fine-tuning, prompt-conditioning, or in-context imitation approaches, not included for comparison? How would MoS-VLA perform relative to them? Comparison with recent state-of-the-art open-sourced VLA models is necessary, including at least diffusion policy, OpenVLA-oft, pi0, pi0.5.

4.How sensitive is MoS-VLA to the number of basis functions, the choice of L1 vs. L2 loss, and the calibration buffer size? Could the authors provide ablation or scaling studies to support these design choices?

I reserve the possibility of raising my score if the authors can adequately address all the concerns raised during the rebuttal phase.

---

> ### Author Response · Authors · 2025-12-03
>
> **1.The real world tests (block lifting, goal reaching, pen insertion) are very short-horizon and low-complexity. These setups mainly assess spatial calibration, not compositional reasoning or long-horizon manipulation, such simple tasks can also be solved by small scale reinforcement learning policy, making it unclear whether the proposed one-shot adaptation method scales to realistic long-horizon or contact-rich tasks, in which the VLA outperform recent tranditional policy learning framework.**
>
> This work studies the domain generalization capabilities of the model, but not the capabilities of the model itself. Any improvement to the model’s long horizon performance is orthogonal to our work. For example, imagine each basis function is a separate copy of OpenVLA or pi0.5. Then their linear combination will clearly perform at least as good as the base model. But in general, we find that domain transfer is very much an open problem. Our experiments show that even the SOTA Pi-0.5 fails to transfer to this new setting without additional training.
>
> **2.Only a single robot platform (Franka Panda) and three real-world tasks are reported, each adapted from demonstration by one expert trajectory. There is no systematic analysis of robustness to in-the-wild scenarios such as noisy demonstrations, viewpoint shifts, or sensor noise, which limits confidence in the generality of the claimed one-shot transfer.**
>
> We run an additional simulated block lifting experiment, but change the camera angle to a side view and the robot morphology to a UR5E robot arm. We find that our approach’s performance does not degrade; It still achieves a 85% success rate over 20 trials.
>
> **The paper compares only to OpenVLA and not to other lightweight adaptation strategies (e.g., LoRA-based finetuning, prompt-conditioning, diffusion-based adapters, or in-context imitation frameworks). Without these, the improvement magnitude is difficult to interpret.**
>
> We compare our method against LoRA finetuning OpenVLA and OpenVLA-OFT. We find that both of these methods require minutes to train whereas our method takes only seconds to calibrate. Furthermore, our method fits on a 3090, whereas both of these methods require too much RAM to do so. Lastly, we also compare against finetuning only the last layer of the OpenVLA model (Linear probing). We find that this approach can fit on a 3090, but performs worse than our approach and takes longer to train. See Appendix F. We also compare against in-context adaptation methods, and find that they fail to generalize in our setting. See Table 1.
>
> **Key hyperparameters such as the number of basis functions, choice of L1 vs. L2 objective, and calibration buffer size are fixed without detailed justification. No experiments isolate how these factors affect adaptation speed or accuracy.**
>
> We have added ablations to these parameters. Please see Appendix E. To summarize the results, the algorithm is stable with respect to the number of basis functions. Increasing the calibration buffer size slightly improves performance at the cost of compute. L1 performs better than L2.
>
> **This method should be framework agnostic, did author attempt using other open-sourced VLA models for testing (e.g., pi0)? Or this method only valid for OpenVLA?**
>
> In general, the theory is architecture agnostic, and so we can apply this method to any VLA. However, pi0 is a special case because it is a flow-matching based model and so its output requires numerous forward passes to generate an action. If you naively applied our current algorithm, you would have to backpropagate through the generative process, which is many forward passes of a billion parameter model. This is infeasible in today’s architecture due to memory constraints. However, in parallel to this work, we are developing a modified version of this algorithm for the diffusion/flow matching setting which requires numerous algorithmic and theoretic advancements. In conclusion, it is possible to apply this method to pi0 but it is beyond the scope of this work.

---

### Official Review · Reviewer_Y2p9 · 2025-10-31

**Soundness:** 3
**Presentation:** 3
**Contribution:** 2
**Rating:** 4
**Confidence:** 3

**Summary:**

The paper proposes a one-shot, gradient-free adaptation method for vision–language–action (VLA) models that substantially reduces computational overhead during online adaptation. The core idea is a function-encoder approach: multiple basis functions are pretrained to span the policy space, and at adaptation time, only the basis coefficients are fine-tuned by solving a linear program. Experiments indicate that the method significantly improves OpenVLA performance on out-of-distribution VLA tasks.

**Strengths:**

First, the paper is clearly written and easy to follow. The paper adopts a function-encoder algorithm with intuitive appeal and theoretical grounding, and scales it to a modern VLA architecture capable of handling high-dimensional vision–language inputs. Finally, the paper provides extensive experimental evaluations, including the real robot experiments.

**Weaknesses:**

- While the paper emphasizes computational efficiency, a quantitative analysis is missing. Comparison against fine-tuning baselines (e.g., LoRA tuning) in both accuracy and efficiency (e.g., wall-clock time, FLOPs, memory/VRAM usage) should be included to support the main claim.

- The paper states that “transformer basis functions” are used, yet the bases appear to differ only in their output heads. According to Appendix C, these are effectively two-layer MLP heads on top of a pretrained VLA transformer’s feature space. Are such two-layer MLP bases sufficient to span the space of robot tasks more broadly? Under what conditions does the approach fail to generalize?

- Although computational overhead scales linearly with the number of bases, achieving adequate expressivity may require many bases. How does performance change as the number of bases increases?

- Evaluating with an L1 loss may be misleading because the action vector comprises distinct components (position, angle, and gripper state). It may also advantage the proposed method, which directly targets L1 minimization. Task-level metrics such as success rate would provide a more comprehensive and fair evaluation.

- Because the proposed approach introduces additional parameters and computation relative to vanilla OpenVLA (e.g., extra LoRA parameters and basis heads), a fair comparison should include a fine-tuned OpenVLA variant with matched parameter counts (same LoRA configuration and head size) and matched optimization budget (e.g., identical numbers of gradient steps).

**Questions:**

Please refer to the weaknesses part.

---

> ### Author Response · Authors · 2025-12-03
>
> **While the paper emphasizes computational efficiency, a quantitative analysis is missing. Comparison against fine-tuning baselines (e.g., LoRA tuning) in both accuracy and efficiency (e.g., wall-clock time, FLOPs, memory/VRAM usage) should be included to support the main claim.**
>
> A key detail in our experiment is that we only use a 3090 at inference time. We are unaware of any other methods that can adapt using such a small overhead. Nonetheless, we run experiments to demonstrate the advantage in terms of compute and memory. See Appendix F. We highlight that our method takes only seconds to calibrate whereas other approaches take minutes. Furthermore, our approach fits on a 3090, whereas LoRA finetuning either OpenVLA or OpenVLA-OFT does not.
>
>
> **Are such two-layer MLP bases sufficient to span the space of robot tasks more broadly?**
>
> Prior works have argued that  independent neural network basis functions can be equivalently represented using one larger neural network by simply stacking the layers and appending zeros when necessary. Empirically, prior work has shown that the difference between independent basis functions and independent output heads is minor, see [4] Appendix F.  They even showed that the shared basis function body method generalizes better to certain out-of-distribution functions. However, changing OpenVLAs output from discrete distributions to a continuous basis requires some changes to the functionality of the main body, and this is why LoRA finetuning is necessary. To summarize, the entire OpenVLA body will play a role in the expressivity of the function space.
>
> **How does performance change as the number of bases increases?**
>
> See appendix E. We find that the performance is largely stable between 8, 16, and 32 basis functions. However, more basis functions take slightly longer to converge, and we posit that this is because it is a more expressive space. Furthermore, 4 basis functions perform worse than the rest, suggesting that the dimensionality of the function space is greater than 4.
>
>
> **Task-level metrics such as success rate would provide a more comprehensive and fair evaluation.**
>
> Please see Table 1, where we compare success rates on both simulated and real world tasks.
>
> **Because the proposed approach introduces additional parameters and computation relative to vanilla OpenVLA (e.g., extra LoRA parameters and basis heads), a fair comparison should include a fine-tuned OpenVLA variant with matched parameter counts (same LoRA configuration and head size) and matched optimization budget (e.g., identical numbers of gradient steps).**
>
> OpenVLA was trained extensively using a massive amount of compute and it fully trained all of the parameters (i.e., not using LoRA). It’s process used 64 A100 for 15 days, and our compute budget is less than 10% additional training time). Consequently, it is unlikely that we can improve the performance of OpenVLA through additional training. We intentionally chose to use the exact same dataset as OpenVLA so that this comparison is fair. In other words, OpenVLA has been trained to convergence on this exact dataset and LoRA tuning is not going to improve its performance.
>
> Additionally, we finetune because we need to change the model from outputting discrete distributions to a set of basis functions, ie to change its behavior. However, LoRA in general does not make a model more expressive; To convince yourself of this, notice that you can multiply the LoRA weights into a model, and recover the exact same model architecture with the original number of parameters. Therefore, its expressivity has not changed.
>
> Lastly, the extra parameters in the basis function heads are extremely small relative to the main LLama2 body, which consists of 7.5 billion parameters. Specifically, OpenVLA has a total of 7,652,065,472 parameters. Our approach has 7,671,122,224 parameters, a 0.2% increase.

---

### Official Review · Reviewer_awN2 · 2025-10-31

**Soundness:** 3
**Presentation:** 2
**Contribution:** 2
**Rating:** 4
**Confidence:** 4

**Summary:**

This paper proposes MoS-VLA, a novel way of adapting vision-language-action (VLA) models via basis functions. The idea of this paper stems from the mixture of experts approach in language models, in which instead of using one action head from OpenVLA, you use multiple action heads and take a linear combination of them to reproduce the expert actions. This has smaller overhead (thanks to parallelization) and can learn a more expressive set of actions by learning both the basis actions (by ensuring orthogonality between each basis) and the action weights. When deployed on a set of simulated and real world manipulation tasks, MoS-VLA is able to distill a useful set of action representations to adapt to a new task with only one demonstration. As a result, MoS-VLA is able to exhibit nonzero success rate in both simulated and real world tasks while the barebone policy cannot complete the task at all.

**Strengths:**

1. I liked the analysis that you have performed across the OXE dataset, which demonstrated the need of many basis functions and the role coefficient space plays.
2. The paper itself flows very naturally. I am able to know most about the method in the first pass, which I haven’t been able to do for many other papers.
3. Both simulated and real world settings are evaluated, which gives reproducible insights into the performance of MoS-VLA.

**Weaknesses:**

1. I believe there are better ways to address some of the design choices and reasons for mentioning terminology. For example, the paper discussed Banach space structure, although I have not seen relevant work that is being cited. I also believe that there are better empirical works that justify why using L1 regression is better, as shown in works such as OpenVLA-OFT [1].
2. Additionally, some of the terms such as action are being ill-defined here, as OpenVLA used discrete action outputs, and from what I can parse here, you have used continuous action outputs.
3. Building upon (2), this might be one of the reasons why you are able to achieve a lower L1 validation error when compared to OpenVLA: if you were to simply decode your actions from output tokens, then it is inherently less expressive than a set of MLPs (which are going to be more parameterized than that of the single action head + detokenizer with vanilla OpenVLA).

Please correct me if there any conceptual misunderstandings from my part.

References:

[1] Kim, M. et al,. 2025. “Fine-Tuning Vision-Language-Action Models: Optimizing Speed and Success”. RSS.

**Questions:**

1. Are there any concerns about overfitting in your method’s setup? Additionally, is it possible to demonstrate that if you have more demonstrations, MoS-VLA’s performance does not degrade?
2. Is there any way to show what each action head is attending to when rolling out a policy? I believe this can be interesting to see the interpretability aspect of the method.
3. How does this method compare against other adaptation methods that are available?

---

> ### Author Response · Authors · 2025-12-03
>
> **The paper discussed Banach space structure, although I have not seen relevant work that is being cited.**
>
> While we are unaware of other works using Banach spaces in a VLA setting, it is common to represent spaces of functions as a Banach space, e.g., see [3]. We will improve the discussion on Banach spaces in the revised manuscript.
>
> **I also believe that there are better empirical works that justify why using L1 regression is better, as shown in works such as OpenVLA-OFT [1].**
>
> Indeed there are other works showing that L1 loss is better for this problem, but there are a few considerations. 1) These methods are partially evaluated using L1 accuracy [1,2], so one may expect L1 is better for training. This may explain why OpenVLA-OFT switches to L1 over token-based methods. 2) Even if L1 is better for a standard VLA model, it is possible that L2 is better for a function encoder-style model. Indeed, all prior works on function encoders use L2 despite the diversity of domains. Nonetheless, our early experiments found that L2 was not effective in this setting. We will mention this connection in the paper, and for the sake of completeness, we include this ablation in the appendix. See Appendix E.
>
> **Some of the terms such as action are being ill-defined here, as OpenVLA used discrete action outputs, and from what I can parse here, you have used continuous action outputs**
>
> We define the action space to be continuous because the action data in RTX is continuous. OpenVLA discretizes this continuous space for simplicity, but this is just a design decision. If you are interested in a discretized version of our algorithm, prior work [4] defines a discrete inner product for classification tasks. Ultimately, we decided to omit the discrete formulation since it is only a design decision of OpenVLA, and discrete definitions are not necessary to understand our work. We will improve the discussion of OpenVLA to include this detail (See Appendix D).
>
> **If you were to simply decode your actions from output tokens, then it is inherently less expressive than a set of MLPs (which are going to be more parameterized than that of the single action head + detokenizer with vanilla OpenVLA)**
>
>
> The main neural body of both methods is a Llama 2 model with 7.5 billion parameters, and therefore the number of parameters is not significantly changed by the output layers. Specifically, OpenVLA has a total of 7,652,065,472 parameters. Our approach has 7,671,122,224 parameters, a 0.2% increase. For a discussion on the expressivity of LoRA, please see our response to reviewer Y2p9. Lastly, it is true discrete distributions and continuous scalars have different inductive biases. For example, consider modeling a function where the true output distribution should be uniform over the set {-1, 1}. While a discrete distribution approximation can match this distribution, a continuous scalar approximation can only output a single value between -1 and 1, likely the mean. We have improved the discussion on parameter count and expressivity in the paper.
>
>
> **Are there any concerns about overfitting in your method’s setup?**
>
> This is an important question. In figure 4, we show that our approach has slightly better performance on the in-distribution validation datasets than OpenVLA, indicating our method has reduced overfitting compared to OpenVLA. For the OOD datasets, you'll notice that both approaches have degraded performance. However, the degradation for our method is much smaller than for OpenVLA. We believe this is due to mixed context overfitting, which our method avoids. See appendix A.
>
> **is it possible to demonstrate that if you have more demonstrations, MoS-VLA’s performance does not degrade?**
>
> We perform ablations on the size of the calibration buffer during training, and find that a larger calibration buffer slightly improves performance. However, this naturally comes at the cost of more compute, so there is a tradeoff here of accuracy vs compute time. We have added this ablation to Appendix E. We also ablate the number of demonstrations at inference time, and find that increasing the number of calibration data points improves performance, again at the cost of compute time. See Appendix F.
>
> **How does this method compare against other adaptation methods that are available?**
>
> We perform additional comparison against Pi-0.5 and ICRT. We find that both of these approaches fail outright in our setting, suggesting that even simple domain transfer is extremely challenging for SOTA models. Another related work, RICL, is not applicable because it requires substantially more data than this setting provides. See new results in Table 1. Additionally, we compare against LoRA finetuning OpenVLA and OpenVLA-OFT. See Appendix F. We find that these methods require minutes to train, whereas ours requires only seconds to calibrate. Furthermore, they require a substantial amount of VRAM and do not fit on a RTX 3090, whereas our method does.

---

### Official Review · Reviewer_1LCB · 2025-11-03

**Soundness:** 3
**Presentation:** 3
**Contribution:** 2
**Rating:** 6
**Confidence:** 3

**Summary:**

The authors propose "Mixture of Skills VLA" (MoS-VLA), a framework that models a robot's policy as a linear combination of a few learned basis functions (or "skills"). These basis functions in practice resemble many different action heads after a base VLM (here, the llama 2 model within OpenVLA). At test time, a expert demonstration from a new task is used to infer the the linear coefficients by solving a L1-norm convex optimization problem (which doesnt require gradient updates since it uses cvxpy). MoS-VLA achieves high success rates on 5 new sim-and-real robot tasks where the 0-shot OpenVLA baseline fails.

**Strengths:**

The idea to use learned basis functions and some simple convex optimization at test time for adaptation is wonderful. It provides a novel approach to adaptation, perhaps most reminiscent of older control theory papers.

The results clearly demonstrate, on some simple tasks, that OpenVLA's struggle to zero shot adapt, can be solved with a little bit of convex optimization and a more suitable architecture.

The t-sne plot visualization of the coefficients is very interesting, and perhaps could be used as a method to measure the quality and diversity of data.

**Weaknesses:**

Experimental evaluation is limited: My first main concern is that, after using many GH200 GPUs for a few hours, the actual evaluation is overly simplified in a couple of ways: (1) very simple block (and one pen) pick and place tasks in the real world, (2) disabling
forward/backward motions. If I remember correctly, openvla can take as input both robot proprioception and multiple camera views but the authors have chosen to only provide one view and no joint angles as input. Both of these can be easily added to the pipeline, especially given the model itself is trained on the hugely diverse RT-X dataset.

Further, it might be helpful to compare with and use recent VLAs like pi0 that should be able to complete simpler tasks zero shot. Perhaps the authors can use test on the tasks like those in [1], where pi0 fails, for a more comprehensive evaluation?

Missing related work with similar (or atleast relevant) high-level ideas: A few highly relevant papers on few-shot adaptation are missing from the related work, and possibly from the baselines. These papers also aim at generalization to new tasks and objects with a few-examples, and gradient-free (in-fact, completely optimization-free) updates. RICL [1] would be the closest work of the following [1,2,3,4] to MoS-VLA.

[1] RICL: Adding In-Context Adaptability to Pre-Trained Vision-Language-Action Models, CoRL 2025

[2] Point Policy: Unifying Observations and Actions with Key Points for Robot Manipulation, CoRL 2025

[3] REGENT: A Retrieval-Augmented Generalist Agent That Can Act In-Context In New Environments, ICLR 2025

[4] Generalization to New Sequential Decision Making Tasks with In-Context Learning, ICML 2024

**Questions:**

Please see weaknesses

---

> ### Author Response · Authors · 2025-12-03
>
> **My first main concern is that, after using many GH200 GPUs for a few hours, the actual evaluation is overly simplified in a couple of ways: (1) very simple block (and one pen) pick and place tasks in the real world**
>
> The tasks are simple because generalizing to new domains is currently very challenging for SOTA models. In new ablations, we evaluate this model against PI-0.5 and ICRT. We find that these approaches also fail even on these relatively simple problems, and hence they would require extensive finetuning. In other words, the problem we are addressing is the issue of domain transfer more than capability of the VLA itself. We have added these results to the paper, see Table 1.
>
> **Openvla can take as input both robot proprioception and multiple camera views but the authors have chosen to only provide one view and no joint angles as input**
>
> OpenVLA itself only takes a single view as input (See [1], page 9, footnote 3). On the other hand, OpenVLA-OFT finetunes the base model on individual domains to take multi-view and proprioceptive states (See [2], page 4). In any case, this is orthogonal to our method.
>
> **Further, it might be helpful to compare with and use recent VLAs like pi0 that should be able to complete simpler tasks zero shot. Perhaps the authors can use test on the tasks like those in [1], where pi0 fails, for a more comprehensive evaluation?**
>
> In new ablations, we compare against Pi-0.5 and ICRT, and find that both of these methods fail out-of-the-box on the tasks from the paper. RICL is not directly comparable because it requires 20 similar tasks and 20 shots each, and so it does not perform under these low data assumptions. We have improved the discussion in the related works and experiments to include these details.

---

### Author Response · Authors · 2025-12-03

**Message to the AC:**

Thank you for taking the time to review our paper. We know that you must have a large workload given present circumstances, and so we have summarized the main concerns of our reviewers and responses in the pages below:

**The on-robot experiments are somewhat simple.**

Our on-robot experiments are somewhat simple because that is where SOTA robot learning is for zero-shot and few-shot domain transfer. To demonstrate this, we run additional experiments with the SOTA Pi-0.5 model, and the in-context ICRT method, and find that they both completely fail in our experiments. See the additions to Table 1. This demonstrates that domain transfer is very much an open problem, and even simple settings are beyond the reach of current methods.

**How does this compare to other few-shot learning methods?**

In addition to the ICRT algorithm mentioned above, we also compare against LoRA finetuning SOTA models such as OpenVLA and OpenVLA-OFT. We find that in terms of loss, our method is similar to these other approaches. However, the true advantage lies in the efficiency. Our method calibrates in only a few seconds, whereas even LoRA finetuning these models takes upwards of 5 minutes. Furthermore, our method uses less than half of the VRAM as these approaches, meaning that our method is deployable even on an RTX 3090, whereas these other methods require much more expensive hardware. The only finetuning approach that fits on a 3090 is to only finetune the last few layers of OpenVLA, but its performance is significantly worse than our method. Please see Appendix F. In summary, our approach achieves similar performance to fine tuning but with much lower overhead.

**How sensitive is this approach to hyper parameters?**

We run additional ablations and present them in Appendix E. We summarize our findings below:
* We train with 4,8,16, and 32 basis functions. We find that performance for 4 basis functions is slightly worse than the others but 8,16, and 32 basis functions perform almost identically. This indicates that the dimensionality of the function space is around 8, and that using extra basis functions beyond the dimensionality of the space does not degrade performance. This aligns with the intuition from functional analysis and prior work [4].
* We ablate the number of examples used to compute the coefficients both during training and inference. During training, we find that increasing the size of the calibration buffer slightly improves performance, but comes at the cost of more compute. During inference, we get a similar result. For example, going from 32 to 512 data points decreases the L1 error by about 10% while increasing the compute time from 2 seconds to 13 seconds. We also verify that the success rate does not degrade either. With 10 expert demonstrations instead of 1 on the simulated block lifting task, the success rate improves to 75%.
* We compare L1 vs L2 distance definitions. We find that L2 performs slightly worse than L1, hence validating our design choice.
Please see Appendix E for more details. In summary, our approach is insensitive to the number of basis functions and scales gracefully as you provide more calibration data.

Please see Appendix E for more details. In summary, our approach is insensitive to the number of basis functions and scales gracefully as you provide more calibration data.

---

> ### Author Response · Authors · 2025-12-03
>
> **How expressive is the function space? Don't the basis function heads share a body?**
>
> The basis function heads do share a body, but prior work (see Appendix F in [4]) has argued that this is equivalent to having k independent basis functions. To illustrate this, suppose you have trained k independent basis function neural networks. If you simply stack the weights of the neural networks, and fill in zeros where necessary, you can recover a single neural network whose output is identical to the k independent basis functions. Consequently, there is an equivalence here, where the body of the neural network is effectively performing different computations for each basis function. In fact, this is why we must LoRA finetune the main body of OpenVLA in our method. We are changing the internal working of the OpenVLA model to support these k basis function heads.
>
> **Do the extra parameters in your approach make it an unfair comparison?**
>
> First, since we reuse the 7 billion parameter OpenVLA body, the number of parameters is largely unchanged. Specifically, OpenVLA has a total of 7,652,065,472 parameters. Our approach has 7,671,122,224 parameters, a 0.2% increase. Second, LoRA finetuning does not make a model more expressive. To convince yourself of this, note that you can multiply the LoRA weights into a model and recover the exact finetuned model as output, but now without LoRA parameters. Thus, the expressivity of the model has not changed. Third, we intentionally use the exact same dataset used to train OpenVLA. OpenVLA itself is trained using 64 A100s for 15 days, and our compute budget is less than 10% additional training time. Thus, OpenVLA has effectively been trained to convergence on the exact dataset we train on as well, whereas we require additional training because we are changing the behavior of the model.
>
> **Final thoughts**
>
> In accordance with the above discussion, we have added the extra experiments to the paper and made changes throughout. We would especially like to highlight the additions to Table 1 and the Appendix sections E and F.
>
> We believe that given a  normal rebuttal process, we would have improved our scores. The original reviews are borderline, but notably have low confidence. Furthermore, some reviewers mentioned they would raise their scores if their concerns were addressed. Given that the results of our ablations and baselines support our arguments, as well as addressing their concerns above,  we believe these reviewers would have raised their scores. We thank you for your time.
>
>
>
> **References:**
>
> [1] Moo Jin Kim, Karl Pertsch, Siddharth Karamcheti, Ted Xiao, Ashwin Balakrishna, Suraj Nair, Rafael Rafailov, Ethan Foster, Grace Lam, Pannag Sanketi, Quan Vuong, Thomas Kollar, Benjamin Burchfiel, Russ Tedrake, Dorsa Sadigh, Sergey Levine, Percy Liang, and Chelsea Finn. Openvla: An open-source vision-language-action model.
>
> [2] Moo Jin Kim, Chelsea Finn, and Percy Liang. Fine-tuning vision-language-action models: Optimizing speed and success.
>
> [3] Oden, J. T. and Demkowicz, L. F. Apply Functional Analysis. CRC Press, third edition edition, 2018.
>
> [4] Tyler Ingebrand, Adam J. Thorpe, and Ufuk Topcu. Function encoders: A principled approach to transfer learning in hilbert spaces

---

### Meta-Review · Area_Chair_KXSy · 2025-12-12

**Summary:**

Real-robot evaluation is too simple / short-horizon; limited environments & platforms: few, simple tasks; one robot; unclear scaling to harder, long-horizon, contact-rich settings [1LCB, qder]

Baseline coverage: need comparisons to Pi-0 / Pi-0.5, ICRT/RICL, LoRA/OFT, prompt/in-context baselines and newer VLAs; fairness of comparisons [1LCB, qder, Y2p9]

Efficiency evidence: missing quantitative wall-clock/FLOPs/VRAM vs fine-tuning; claim of “seconds vs minutes” needs valuation [Y2p9]

Hyperparameters & scaling: number of bases (k), calibration buffer size (train/infer), L1 vs L2 objective; sensitivity & trade-offs [1LCB, qder, awN2]

Expressivity / architecture: do shared-body basis heads truly span a rich function space; is this just extra heads; discrete vs continuous action definition; parameter-count fairness; is LoRA changing expressivity [awN2, Y2p9]

Input design choices: single camera view, no proprio; disabling fwd/back motion; whether this handicaps baselines or limits conclusions [1LCB]

Robustness & generality tests: behavior under noisy demos, viewpoint shifts, lighting/sensor noise; other robots/datasets; framework-agnostic applicability (e.g., Pi-0) [qder, 1LCB]

Positioning / related work & theory: missing/under-cited related few-shot & in-context works (RICL, REGENT, etc.); Banach-space discussion [1LCB, awN2]

**Reviewer Concerns:**

Baselines expanded & context: Addressed. Added Pi-0.5 and ICRT (reporting failures out-of-the-box); compared to LoRA fine-tuning of OpenVLA/OpenVLA-OFT, and last-layer (linear probe); discussed RICL data requirements [1LCB, qder, Y2p9]

Efficiency measurements: Addressed. Reported seconds to calibrate vs minutes to fine-tune; fits on RTX 3090 vs fine-tuning not fitting; Appendix F for wall-clock and VRAM  [Y2p9]

Hyperparameter ablations: Addressed. k = 4/8/16/32 (≥8 all similar; 4 worse), buffer size (bigger = small gains, more compute), L1 > L2 at train/infer; reported latency trade-offs (e.g., 32→512 pts: −10% L1 error, 2 s→13 s) [1LCB, qder, awN2]

Expressivity / architecture: Sufficiently Addressed. Argued shared-body heads are equivalent to independent bases (stacking argument); LoRA needed to change OpenVLA’s body to support bases; LoRA doesn’t increase expressivity; extra params ≈ +0.2% (7.652B → 7.671B). Clarified continuous action choice vs OpenVLA’s discrete tokens; added L1 vs L2 rationale [awN2, Y2p9]

Robustness / generality: Partially Addressed. Added UR5e sim with side-view camera); clarified OpenVLA single-view input and OFT’s multi-view/proprio being orthogonal. Viewpoint & robot morphology shown; no explicit tests with noisy demos/lighting variations in real-world [qder, 1LCB]

Real-robot simplicity: Partially addressed. Justified simplicity by showing Pi-0.5 & ICRT fail even there (Table 1); framed the focus as domain transfer rather than long-horizon capability. Motivation shown, but tasks remain simple [1LCB, qder]

Input design / motion constraints: Not Addressed. Clarified OpenVLA’s single-view and OFT differences; did not directly address “disabling forward/backward motions.” [1LCB]

Related work & theory: Addressed. Added/acknowledged RICL/REGENT/etc. context; promised to strengthen Banach-space discussion; linked to function-encoder literature [1LCB, awN2]

Framework-agnosticity: Partially Addressed. Explained why Pi-0/flow-matching needs many passes, so current method impractical. Rationale is given, but no experiments [qder]

Interpretability of heads: Not Addressed. Request to show what each head attends to... no analyses/visuals [awN2]

Parameter-matched fairness experiment: Partially Addressed. Reviewer asked for matched-param LoRA OpenVLA with same budget; authors argued OpenVLA converged & LoRA won’t help; no matched run provided. No empirical underpinning[Y2p9]

**Reviewer Scores:**

The original scores were 6, 4, 4, 4. As to how reviewers would have changed their position, I cannot guess. Although the authors did address several of the reviewers' concerns in the discussion, several aspects remained in the balance or were not countered satisfactorily.

---

### Decision · Program_Chairs · 2026-01-26

Reject